# *YARS2* Missense Variant in Belgian Shepherd Dogs with Cardiomyopathy and Juvenile Mortality

**DOI:** 10.3390/genes11030313

**Published:** 2020-03-14

**Authors:** Corinne Gurtner, Petra Hug, Miriam Kleiter, Kernt Köhler, Elisabeth Dietschi, Vidhya Jagannathan, Tosso Leeb

**Affiliations:** 1Institute of Animal Pathology, Vetsuisse Faculty, University of Bern, 3001 Bern, Switzerland; corinne.gurtner@vetsuisse.unibe.ch; 2Institute of Genetics, Vetsuisse Faculty, University of Bern, 3001 Bern, Switzerland; petrahug@bluewin.ch (P.H.); elisabeth.dietschi@vetsuisse.unibe.ch (E.D.); vidhya.jagannathan@vetsuisse.unibe.ch (V.J.); 3Department/Hospital for Companion Animals and Horses, University Clinic for Small Animals, Internal Medicine Small Animals, University of Veterinary Medicine, 1210 Vienna, Austria; Miriam.Kleiter@vetmeduni.ac.at; 4Institute of Veterinary Pathology, Justus-Liebig-University Giessen, 35392 Giessen, Germany; kernt.koehler@vetmed.uni-giessen.de

**Keywords:** *Canis lupus familiaris*, whole genome sequence, animal model, precision medicine, mitochondrium, translation, Groenendael, Laekenois, Malinois, Tervueren

## Abstract

Dog puppy loss by the age of six to eight weeks after normal development is relatively uncommon. Necropsy findings in two spontaneously deceased Belgian Shepherd puppies indicated an abnormal accumulation of material in several organs. A third deceased puppy exhibited mild signs of an inflammation in the central nervous system and an enteritis. The puppies were closely related, raising the suspicion of a genetic cause. Pedigree analysis suggested a monogenic autosomal recessive inheritance. Combined linkage and homozygosity mapping assigned the most likely position of a potential genetic defect to 13 genome segments totaling 82 Mb. The genome of an affected puppy was sequenced and compared to 645 control genomes. Three private protein changing variants were found in the linked and homozygous regions. Targeted genotyping in 96 Belgian Shepherd dogs excluded two of these variants. The remaining variant, *YARS2*:1054G>A or p.Glu352Lys, was perfectly associated with the phenotype in a cohort of 474 Belgian Shepherd dogs. *YARS2* encodes the mitochondrial tyrosyl-tRNA synthetase 2 and the predicted amino acid change replaces a negatively charged and evolutionary conserved glutamate at the surface of the tRNA binding domain of YARS2 with a positively charged lysine. Human patients with loss-of-function variants in *YARS2* suffer from myopathy, lactic acidosis, and sideroblastic anemia 2, a disease with clinical similarities to the phenotype of the studied dogs. The carrier frequency was 27.2% in the tested Belgian Shepherd dogs. Our data suggest *YARS2*:1054G>A as the candidate causative variant for the observed juvenile mortality.

## 1. Introduction

Increased puppy loss is an animal welfare concern associated with emotional and financial burdens to dog breeders. In dogs, the rate of stillbirths and neonatal death within the first weeks of life is known to be high [1,2]. During the first days of life, the death rate in puppies is highest [3] as there are various predisposing factors that can lead to life-threatening illnesses and subsequent death. Non-infectious causes, such as prolonged delivery and dystocia, are the most common factors leading to early death in new-born puppies [3]. Furthermore, hypoxia and respiratory distress syndrome, hypoglycemia, hypothermia, and dehydration can lead to fatal diseases, or predispose for secondary infectious diseases [3]. Structural or functional congenital malformations as further non-infectious causes may lead to early puppy death or be an indication for euthanasia [4]. Various infections may also result in puppy loss [5,6] and bacterial infections in puppies are the most common infectious cause for diseases and overall the second most important cause of mortality in neonatal dogs [7]. The sudden death of puppies after several weeks of normal development is less common as they have survived potentially threatening immunological and nutritional issues associated with birth [1,2].

In Switzerland and Germany, Belgian Shepherd breeders noticed three puppies of the Malinois variety dying from unknown causes with unspecific clinical signs at 6–8 weeks of age. The animals were submitted for post-mortem examinations. The goal of this study was to characterize the phenotype and identify the underlying causative genetic defect.

## 2. Materials and Methods 

### 2.1. Ethics Statement

All dogs in this study were privately owned and samples were collected with the consent of their owners. The collection of blood samples was approved by the “Cantonal Committee For Animal Experiments” (Canton of Bern; permits 75/16 and 71/19).

### 2.2. Breed Nomenclature

The Federation Cynologique Internationale (FCI) describes the Malinois, together with the Groenendael, the Laekenois, and the Tervueren, as varieties of the Belgian Shepherd dog breed. The American Kennel Club, however, officially recognizes the Belgian Malinois, the Belgian Sheepdog (FCI: Groenendael), the Belgian Laekenois (FCI: Laekenois), and the Belgian Tervuren (FCI: Tervueren) as four distinct breeds. In this paper, all references to the breed nomenclature correspond to the FCI standards.

### 2.3. Animal Selection

This study included 474 Belgian Shepherd dogs. Three Malinois puppies, two of them being full siblings, died within the first 8 weeks of life without an obvious clinical cause of death. The remaining dogs represented population controls without consistent phenotypic evaluation. We also used samples of 534 dogs from 72 genetically diverse other breeds, which had been donated to the Vetsuisse Biobank (Appendix A).

### 2.4. DNA Extraction

Genomic DNA was isolated from EDTA blood with the Maxwell RSC Whole Blood Kit using a Maxwell RSC instrument (Promega, Dübendorf, Switzerland).

### 2.5. Linkage and Homozygosity Mapping

Five dogs from one Malinois family consisting of the unaffected parents, two affected, and one unaffected offspring were genotyped on the Illumina canine_HD BeadChip containing 220,853 markers (Neogen, Lincoln, NE, USA). The raw single nucleotide variant (SNV) genotypes are available in File S1. The genotype data were used for a parametric linkage analysis. Using PLINK v 1.09 [8], markers located on the sex chromosomes, with missing genotypes in any of the five dogs, containing Mendel errors, or having a minor allele frequency < 0.05 were excluded. The final dataset contained 93,179 markers. For parametric linkage analysis, an autosomal recessive inheritance model with full penetrance, a disease allele frequency of 0.4, and the Merlin software [9] were applied.

For homozygosity mapping, the genotype data of the two affected full siblings were used. The options --homozyg and --homozyg-group in PLINK were used to search for extended regions of homozygosity > 1 Mb with simultaneous allele sharing. The output intervals were matched against the intervals from linkage analysis in Excel spreadsheets to find overlapping regions. The output of the linkage and homozygosity mapping is given in Appendix A.

### 2.6. Whole Genome Sequencing of an Affected Malinois

An Illumina TruSeq PCR-free DNA library with a 350-bp insert size from one of the two affected Malinois full-siblings was prepared (MA471). We collected 239 million 2 x 150 bp paired-end reads on a NovaSeq 6000 instrument (26.2x coverage). Mapping and alignment were performed as described [10]. The sequence data were deposited under study accession PRJEB16012 and sample accession SAMEA6249500 at the European Nucleotide Archive.

### 2.7. Variant Calling and Filtering

Variant calling was performed as described [10]. To predict the functional effects of the called variants, SnpEff [11] software together with NCBI annotation release 105 for the CanFam 3.1 genome reference assembly was used. We used a vcf-file containing 648 dog and 8 wolf genomes, which were either publicly available [12,13] or produced during other projects of our group [10]. We excluded 10 genome sequences from other Belgian Shepherd dogs from filtering to minimize the risk of having any unrecognized carriers in the control cohort. We employed a hard filtering approach to identify variants at which the affected dog was homozygous for the alternate allele (1/1) while 645 control genomes were either homozygous for the reference allele (0/0) or had a missing genotype call (./.). The accession numbers of all genome sequences used for the analysis are compiled in Appendix A.

### 2.8. Gene Analysis

The dog CanFam 3.1 reference genome assembly was used for all analyses. Numbering within the canine *YARS2* gene corresponds to the NCBI RefSeq accessions XM_543740.6 (mRNA) and XP_543740.1 (protein).

### 2.9. Sanger Sequencing

The *YARS2*:c.1054G>A variant was genotyped by direct Sanger sequencing of PCR amplicons. A 300 bp PCR product was amplified in a 10 µL reaction volume from genomic DNA using AmpliTaqGold360Mastermix (Thermo Fisher Scientific, Waltham, MA, USA) together with primers 5‘-GGT AAG GGA GTT GGA GAC AAA A-3‘ (Primer F) and 5’- GTG ACA AGA TAA CCC GAA TGA A-3’ (Primer R). After an initial denaturation of 10 min at 95 °C, the reaction was incubated for 30 cycles of 30 s at 95 °C, 30 s at 60 °C and 60 s at 72 °C. At the end of the program, a final extension step of 7 min at 72 °C was performed. After treatment with exonuclease I and alkaline phosphatase, amplicons were sequenced on an ABI 3730 DNA Analyzer (Thermo Fisher Scientific). Sanger sequences were analyzed using the Sequencher 5.1 software (GeneCodes, Ann Arbor, MI, USA).

## 3. Results

### 3.1. Phenotype Description

Two 8-week-old full siblings (one male and one female) were both clinically examined because of vomiting and dyspnea. Radiologically, hypoplasia of the trachea was suspected in both animals. Furthermore, the male sibling appeared to have an enlarged heart. Both puppies died suddenly and were submitted for postmortem examination. Necropsy findings in both animals revealed a very pale and enlarged heart. In light microscopy, the cardiomyocytes were swollen with a perinuclear clearing while the rest of the sarcoplasm was displaced and appeared grainy compared to heart tissue from other canine puppies (Figure 1). There was a minimal chronic lymphocytic myocarditis as a result of the degeneration of affected cardiomyocytes. Few cells of the exocrine pancreas had optically empty vacuoles in the cytoplasm, and hepatocytes in both animals had a finely granular appearance. Despite attempting various histologic staining procedures, the accumulated material could not be further characterized. Further testing for canine parvovirosis was negative. The cause of disease and death were likely due to the degenerative changes in the heart, leading to myocardial failure.

A third male puppy became reluctant to move while twitching and showing tremors at six weeks of age. During these bouts of tremors and twitching it exhibited spontaneous urination and defecation. The puppy was treated for myelitis and died suddenly at 8 weeks of age. The necropsy and histologic examination revealed similar changes in the heart as with the other two puppies where cardiomyocytes appeared to have a grainy sarcoplasm. Furthermore, there were mild changes in the brain and spinal cord with few chromatolytic neurons in the spinal cord, mild gliosis in the cerebellum, and a mild mixed cell infiltration of the meninges of the cerebellum. Additionally, the puppy had a catarrhal enteritis, for which the cause could not be determined. The cause of disease remained undetermined after the necropsy.

### 3.2. Genetic Analysis

The pedigree of the three affected puppies was suggestive for a monogenic autosomal recessive mode of inheritance (Figure 2). Linkage analysis of a family with two affected and one unaffected offspring revealed 33 linked genome segments comprising 262 Mb in total. We also performed homozygosity mapping in the two affected siblings. They shared 32 homozygous regions across the genome. The intersection of the linked and homozygous intervals consisted of 13 genome segments totaling 82 Mb (Appendix A).

We sequenced the genome of the affected puppy MA471 at 26.2x coverage. We called SNVs and short indels with respect to the CanFam3.1 reference genome assembly. We then compared these variants to whole genome sequence data of 8 wolves and 637 control dogs from genetically diverse breeds (Table 1).

Only three private protein-changing variants were located in the linked and homozygous regions (Table 2). We confirmed these three variants by Sanger sequencing and genotyped them in a cohort of 96 Belgian Shepherd dogs. For two of the variants we observed all three genotypes in healthy dogs, which excluded them as being causative for a lethal phenotype.

This left only one candidate causative variant, Chr27:16,157,324A>G, for which we did not find any homozygous mutant control dogs. It represented a missense variant in the third exon of the *YARS2* gene encoding tyrosyl-tRNA synthetase 2, which is the mitochondrial tyrosyl-tRNA synthetase (Figure 3). The variant, XM_543740.6:c.1054G>A, is predicted to change a highly conserved glutamate to a lysine, XP_543740.1:p.(Glu352Lys). *In silico* analysis predicted the functional effect of p.Glu352Lys as disease causing, with an 84% probability (PMut, [14]), or to affect protein function, with a score of 0.01 (Sift, [15]).

We genotyped additional dogs for the *YARS2*:c.1054G>A variant to end up with a final cohort of 474 Belgian Shepherd dogs and 534 dogs of genetically diverse other breeds. The genotypes were perfectly associated with the juvenile lethality phenotype (Table 3). All 3 deceased puppies were homozygous mutant while none of the 1000 control dogs carried the mutant allele in a homozygous state. The mutant allele was only detected in Belgian Shepherd dogs but not in any dogs from other breeds. It was most commonly seen in the Malinois variety of Belgian Shepherd dogs, but it also occurred in the Groenendael and Tervueren varieties.

The overall mutant allele frequency in the genotyped Belgian Shepherd cohort was 14.1%. The observed mutant allele frequencies in Malinois (n = 383), Groenendael (n = 36), Tervueren (n = 52), and Laekenois (n = 3) were 0.16, 0.03, 0.07, and 0, respectively.

The genotyped cohort did not represent a truly random representation of the breed as it contained ~20 close relatives to the three cases that were specifically collected for the project. Therefore, the observed carrier frequency of 27.2% in the controls of our Belgian Shepherd cohort may represent an overestimation of the true carrier frequency in the Middle European Belgian Shepherd population. The observed genotype frequencies deviated from Hardy–Weinberg equilibrium with a significant underrepresentation of homozygous mutant dogs (*p* = 2.6 × 10^−4^).

## 4. Discussion

In this study, we identified the *YARS2*:c.1054G>A variant and a mitochondrial translation defect as the potential cause for juvenile mortality in Belgian Shepherd dogs. Protein translation requires aminoacyl-tRNAs to deliver the amino acids that are incorporated into nascent peptide chains. Aminoacylation of tRNAs is catalyzed by a set of aminoacyl-tRNA synthetases, which have highly specific recognition sites for their respective amino acids and the corresponding tRNA molecules.

Eukaryotes have two tyrosyl-tRNA synthetases, the cytoplasmic tyrosyl-tRNA synthetase 1 (YARS1) and the mitochondrial tyrosyl-tRNA synthetase 2 (YARS2) [19]. The three-dimensional structure of human YARS2 has been resolved at high resolution by X-ray crystallography [16,17]. It is composed of a catalytic domain containing the binding sites for ATP and tyrosine and a tRNA binding domain, which mediates the recognition of the mitochondrial tRNA^Tyr^ [16,18]. Similar to other tRNA synthetases, YARS2 is highly conserved throughout eukaryotes. The identified p.Glu352Lys variant replaces a negatively charged carboxyl group of a glutamate with a positively charged amino group of a lysine at the surface of the tRNA-binding domain. Glu-352 is conserved across highly diverse eukaryotes ranging from animals, plants, and fungi down to *Saccharomyces cerevisiae* (Figure 3B). We found only two exceptions, one of them being *Drosophila*, in which Glu-352 is conservatively replaced by the negatively charged aspartate. The other exception is *Schizosaccharomyces pombe*, which has a neutral asparagine at the homologous position. The high sequence conservation of this region suggests that it is functionally important and supports the hypothesis that p.Glu352Lys might affect YARS2 function, most likely by hampering the correct recognition and binding of the mitochondrial tRNA^Tyr^.

In human patients, *YARS2* variants cause a phenotype termed myopathy, lactic acidosis, and sideroblastic anemia 2 (MLASA2; OMIM#613561). This disorder of the mitochondrial respiratory chain shows remarkable variation in the clinical phenotype [20,21,22,23,24,25]. Some patients have a severe multisystem disorder from infancy, including hypertrophic cardiomyopathy and respiratory insufficiency resulting in early death. Other patients only present in the second or third decade of life with sideroblastic anemia and mild muscle weakness [21,22]. Most reported human patients required blood transfusions due to metabolic impairment of erythropoiesis and resulting sideroblastic anemia. Human patients may additionally show lethargy and progressive exercise intolerance; some of them become unable to walk [22]. Patients were further reported to suffer from intermittent vomiting and gastrointestinal problems [24]. Very severe human cases involved additional brain abnormalities [21,22] and proximal renal tubulopathies [22]. The large variability in clinical symptoms seen in human MLASA2 patients overlaps with several of the clinical and pathological findings observed in the deceased Belgian Shepherd puppies, such as the intermittent vomiting, enteritis, and most importantly the cardiomyopathy. The affected puppies did not show clinical signs of a sideroblastic anemia. However, no blood smears were analyzed, which are needed to diagnose sideroblastic anemia in dogs.

The observed strong genotype–phenotype association and the knowledge on the functional effect of YARS2 variants from human MLASA2 patients suggest *YARS2*:c.1054G>A as the candidate causative variant in Belgian Shepherd dogs. Unfortunately, an experimental follow-up analysis on the transcript or protein level to provide functional proof for this hypothesis could not be performed as no suitable tissue samples from the affected dogs were available.

Our studied cohort of Belgian Shepherd dogs had a very high frequency of the mutant allele. The high carrier frequency of 27.2% in unaffected Belgian Shepherd dogs is expected to have resulted in the birth of many affected puppies. Due to the early lethality and the unspecific clinical signs, this recessive genetic defect may have been overlooked for some time. We think that our results warrant the introduction of genetic testing and a breeding program minimizing the risk of producing affected puppies. Future matings should be planned with at least one of the breeding animals being clear (*wt/wt*) to avoid the birth of further homozygous mutant offspring. At the same time, it is important to stress that carriers should not be immediately excluded from breeding. We recommend aiming at a gradual reduction of the mutant allele. An abrupt exclusion of all carrier animals from breeding would lead to a substantial loss of genetic diversity in the breed and a further increase in inbreeding. This in turn is likely to result in the increase of other yet unknown recessively inherited defects. We previously gave the same breeding recommendation for another autosomal recessive genetic defect in dogs [26].

## 5. Conclusions

We identified the *YARS2*:c.1054G>A missense variant as the potential cause for juvenile mortality in Belgian Shepherd dogs. The phenotype of affected puppies is inherited as an autosomal recessive trait and shows parallels to human MLASA2 patients. Our data facilitate genetic testing of Belgian Shepherd dogs to prevent the non-intentional breeding of further affected puppies.

## Figures and Tables

**Figure 1 genes-11-00313-f001:**
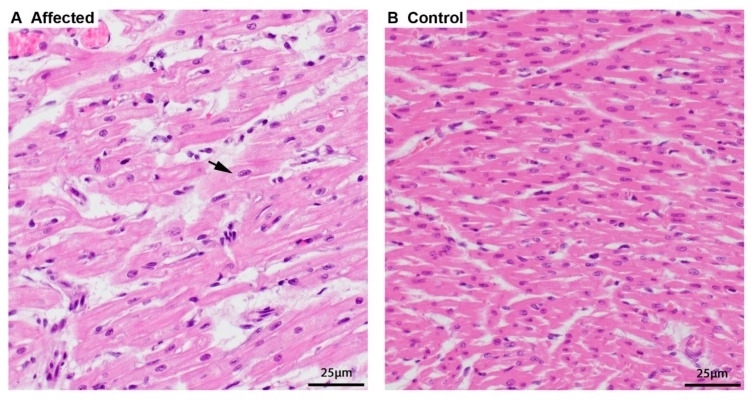
Histopathology of the heart. (**A**) Cardiomyocytes of an affected puppy are swollen and pale, and the sarcoplasm around the nucleus is dispersed (arrow) by finely granular material, hematoxylin and eosin (HE) stain. (**B**) Normal, smaller, and more intense staining of cardiomyocytes of a 5-week-old control puppy, HE stain.

**Figure 2 genes-11-00313-f002:**
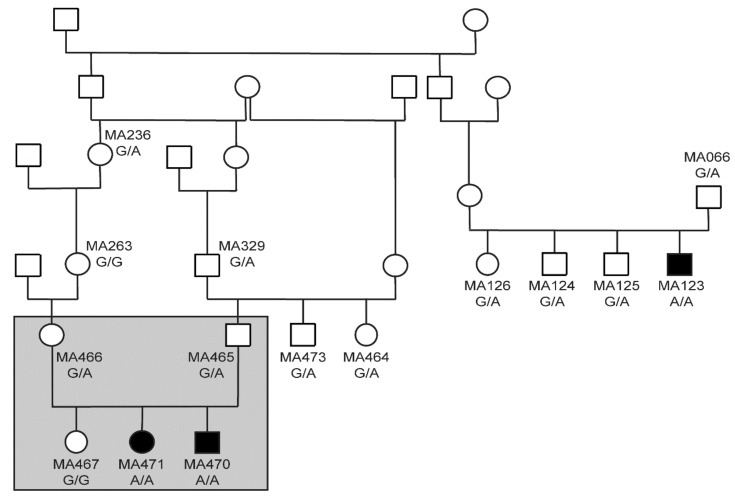
Pedigree of the three investigated cases. Filled symbols represent puppies deceased by the age of six to eight weeks. Open symbols represent unaffected animals. Genotypes at *YARS2*:c1054G>A are indicated for all dogs, from which a DNA sample was available. The grey rectangle indicates the five dogs included in linkage and homozygosity analysis.

**Figure 3 genes-11-00313-f003:**
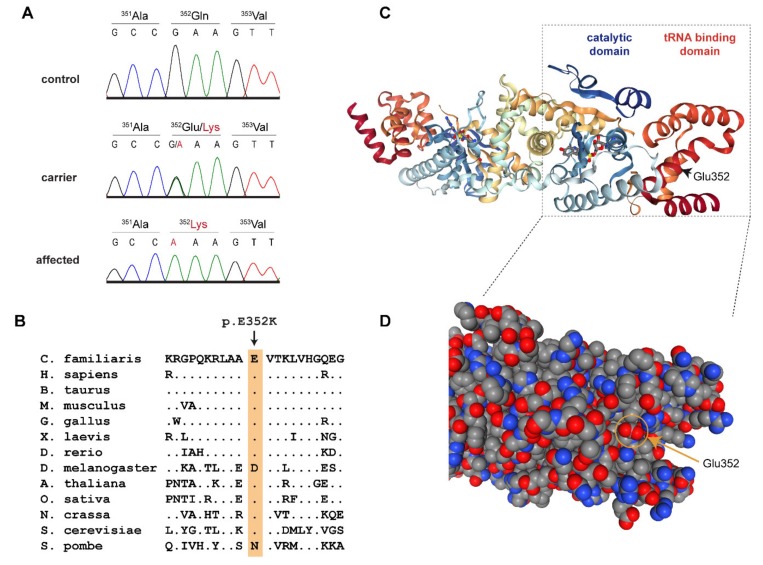
Details of the *YARS2*:p.Glu352Lys variant. (**A**) Representative Sanger electropherograms of a control, a carrier, and an affected dog illustrate the c.1054G>A variant. The amino acid translations are shown. (**B**) Evolutionary conservation. The glutamate at position 352 is highly conserved across all eukaryotes. (**C**) Three-dimensional structure of the human YARS2 protein in complex with an adenylate homolog [16,17,18]. The position of Glu352 is indicated in the right monomer. (**D**) Enlarged details of the structure in a space-filling model. The negatively charged carboxyl group of Glu352 is located at the surface of the tRNA binding domain.

**Table 1 genes-11-00313-t001:** Variants detected by whole genome re-sequencing of an affected puppy.

Filtering Step	Variants
Homozygous variants in whole genome	2,757,566
Private homozygous variants (absent from 645 control genomes) in whole genome	1321
Private homozygous variants in 82 Mb critical intervals	150
Protein-changing private variants in critical intervals	3

**Table 2 genes-11-00313-t002:** Details of three private protein-changing candidate variants.

Chr.	Position	Ref.	Alt.	Gene	HGVS-c	HGVS-p
5	41,681,870	G	A	*SREBF1*	c.1678G>A	p.Gly560Ser
27	3,744,738	C	T	*LOC106557897*	c.443G>A	p.Arg148His
27	16,157,324	G	A	*YARS2*	c.1054G>A	p.Glu352Lys

**Table 3 genes-11-00313-t003:** Genotype phenotype association of the *YARS2*:c.1054G>A variant.

Dogs	A/A	G/A	G/G
Belgian Shepherd cases (n = 3)	3	0	0
Belgian Shepherd controls (n = 471)	0	128	343
Control dogs other breeds (n = 534)	0	0	534

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
