# Peer review of "YARS2 Missense Variant in Belgian Shepherd Dogs with Cardiomyopathy and Juvenile Mortality"

_genes, 2020, doi:10.3390/genes11030313_

Round 1

Reviewer 1 Report

Main comments:

Line 17 - what type of material? Did not find further explanation in the necropsy description under the results section.

Line 78 - as per journal guidelines, please define the "SNV" abbreviation as this is the first time it is used in the text.

Line 79 - please clarify what the authors mean by non-informative markers. Is this referring to markers for which all five dogs carried the same genotype?

Line 90 - The results section states which puppy was sequenced, but I believe it should be also be stated in the methods that one of the affected full siblings was sequenced.

In lines 98-101 the authors state that 645 genomes were used for filtering variants and that genomes from Belgian Shepherd dogs were excluded from analysis. They subsequently state that the accession numbers of all genome sequences used for analysis are compiled in Table S3. However, Table S3 contains 656 listed genomes, 11 of which are from Belgian Shepherd dogs (which they said were excluded from analysis). I suggest rewording this section for clarity.

Line 111 - please include information about the PCR protocol used  

Line 182 - based on how the results were presented up until this point in the paper, this sentence seems to suggest that 474 additional Belgian Shepherd dogs were genotyped for the YARS2 variant. However, Table 3 clearly shows that the total number of Belgian Shepherd dogs genotyped in this study was 474 (which included the 3 affected dogs and the 96 dogs genotyped for the 3 protein-coding variants from whole genome sequencing of the affected puppy). Therefore, I suggest rewording this sentence to state that overall 474 Belgian Shepherd dogs were genotyped for the YARS2 variant, or that 375 additional Belgian Shepherd dogs and 534 dogs from other breeds were genotyped. 

Line 188 - the authors point out that the mutant allele was most commonly seen in the Malinois, but was also seen in the Groenendal and Tervueren varieties. Please include the frequency of the mutant allele in each of the varieties.

Line 191 - the authors' concern that the ~20 close relatives could be significantly affecting the carrier frequency observed seems valid. I suggest calculating the carrier frequency after removing the close relatives for comparison. 

Line 222 - did the puppies have anemia? Do the authors have any theory to explain the highly variable phenotype in humans and dogs?

Lines 240 and 249 - While the results presented in the paper support the authors' conclusions, they have been careful to state throughout the paper that the identified YARS2 variant could be a potential cause for early mortality in the Belgian Shepherd dogs. It might be premature at this point to recommend genetic testing, especially given the very valid points made by the authors at the end of the discussion regarding the risks of increasing inbreeding and reducing genetic diversity should a genetic test be misused to eliminate all carriers (lines 242-247). 

Author Response

(1)

Line 17 - what type of material? Did not find further explanation in the necropsy description under the results section.

Response: Unfortunately, the material could not be further characterized as the additional staining and reactions such as PAS, PAS and Diastase, PTAH, Lipofuscin staining were inconclusive or negative. To examine the accumulated intracellular material, ultrastructural investigation by electron microscopy would have to be performed, which in our cases was not possible. Material used for ultrastructural examinations needs to be fresh and both puppies had died a day or longer before the post-mortem was performed. With the third puppy there was no wet (formalin-fixed but not embedded) material left to perform such an investigation.

To address this issue, we added the following sentence in lines 131 – 132: Despite attempting various histologic staining procedures, the accumulated material could not be further characterized.

(2)

Line 78 - as per journal guidelines, please define the "SNV" abbreviation as this is the first time it is used in the text.

Response: Revised accordingly.

(3)

Line 79 - please clarify what the authors mean by non-informative markers. Is this referring to markers for which all five dogs carried the same genotype?

Response: Non-informative markers are markers that are fixed for one allele in all genotyped animals. We deleted this particular phrase as we also applied a filter for minor allele frequency (MAF ≥ 0.05), which will also remove non-informative markers.

(4)

Line 90 - The results section states which puppy was sequenced, but I believe it should be also be stated in the methods that one of the affected full siblings was sequenced.

Response: Revised accordingly.

(5)

In lines 98-101 the authors state that 645 genomes were used for filtering variants and that genomes from Belgian Shepherd dogs were excluded from analysis. They subsequently state that the accession numbers of all genome sequences used for analysis are compiled in Table S3. However, Table S3 contains 656 listed genomes, 11 of which are from Belgian Shepherd dogs (which they said were excluded from analysis). I suggest rewording this section for clarity.

Response: We revised and expanded this section.

(6)

Line 111 - please include information about the PCR protocol used 

Response: We added the information on the details of the PCR protocol.

(7)

Line 182 - based on how the results were presented up until this point in the paper, this sentence seems to suggest that 474 additional Belgian Shepherd dogs were genotyped for the YARS2 variant. However, Table 3 clearly shows that the total number of Belgian Shepherd dogs genotyped in this study was 474 (which included the 3 affected dogs and the 96 dogs genotyped for the 3 protein-coding variants from whole genome sequencing of the affected puppy). Therefore, I suggest rewording this sentence to state that overall 474 Belgian Shepherd dogs were genotyped for the YARS2 variant, or that 375 additional Belgian Shepherd dogs and 534 dogs from other breeds were genotyped.

Response: We revised this sentence to avoid any misunderstandings. The sentence now reads: “We genotyped additional dogs for the YARS2:c.1054G>A variant to end up with a final cohort of 474 Belgian Shepherd dogs and 534 dogs of genetically diverse other breeds.”

(8)

Line 188 - the authors point out that the mutant allele was most commonly seen in the Malinois, but was also seen in the Groenendal and Tervueren varieties. Please include the frequency of the mutant allele in each of the varieties.

Response: We added the mutant allele frequencies in each of the four varieties to the results section.

(9)

Line 191 - the authors' concern that the ~20 close relatives could be significantly affecting the carrier frequency observed seems valid. I suggest calculating the carrier frequency after removing the close relatives for comparison.

Response: We appreciate the reviewer’s comment. However, as we do not have full pedigree information on each dog in the study, it is impossible to exactly specify a truly representative carrier frequency. Even, if 20 carriers are arbitrarily removed from the cohort, the carrier frequency would only drop from 27.2% to 23.9%. Thus, the effect of the relatively small number of close relatives on the overall carrier frequency is not dramatic.

(10)

Line 222 - did the puppies have anemia? Do the authors have any theory to explain the highly variable phenotype in humans and dogs?

Response: According to the treating clinicians none of the three puppies showed any signs of anemia and cytological blood smears were therefore not analyzed. Such a cytological blood smear is needed to diagnose sideroblastic anemia in dogs. The bone marrow of the two Swiss sibling puppies appeared within normal limits during histology. As we do not know if the puppies suffered from sideroblastic anemia, we can unfortunately not address this point.

We added the following text to the discussion: “The affected puppies did not show clinical signs of a sideroblastic anemia. However, no blood smears were analyzed, which are needed to diagnose sideroblastic anemia in dogs.”

(11)

Lines 240 and 249 - While the results presented in the paper support the authors' conclusions, they have been careful to state throughout the paper that the identified YARS2 variant could be a potential cause for early mortality in the Belgian Shepherd dogs. It might be premature at this point to recommend genetic testing, especially given the very valid points made by the authors at the end of the discussion regarding the risks of increasing inbreeding and reducing genetic diversity should a genetic test be misused to eliminate all carriers (lines 242-247).

Response: This is indeed a delicate question. On the one hand, the causality of the YARS2:c.1054G>A variant is not conclusively proven. On the other hand, there is a lethal phenotype and a potentially very high carrier frequency in the population.

We very carefully thought on the exact wording of our breeding recommendation. It currently reads: “We think that our results warrant the introduction of genetic testing …”

This wording should clearly indicate that this is our subjective recommendation. We therefore did not change the text with respect to the reviewer’s comment.

Reviewer 2 Report

The manuscript included describes a phenotype characterized by sudden onset of respiratory difficulty, digestive concerns, and lethargy, followed by death at around 8 weeks of age. Clinical examination of affected puppies noted tracheal and cardiac anomalies from radiographs, and postmortem histology identified cardiomyocyte anomalies, myocarditis, and in one case, brain and spinal cord changes. Pedigree analysis suggested autosomal recessive inheritance likely. No obvious candidate genes were noted that could explain the collection of clinical signs; linkage and whole genome sequencing identified genomic regions and variants to assess for pathogenicity. One variant segregated within the family and is in a conserved residue. Interestingly, there is a very high carrier rate for this variant, yet this phenotype has not been characterized before. 

I have a couple questions and minor edits only, and feel this is a strong manuscript. 

  1. The sentence spanning lines 41-43 needs some attention. "fatal diseases" and "puppy loss" seem redundant, as "fatal" would indicate loss of puppy. I would suggest the sentence read "...lead to fatal diseases, or predispose for secondary infectious diseases [3]." As written with the "and" suggests secondary diseases could occur after puppy loss.
  2. I would like line 50 to indicate the number of dogs cited, rather than "several" - is this sentence referring to the three affected dogs included in the study, or are there more families with puppy loss?
  3. I was not aware the Belgian Shepherd was described as 4 AKC breeds and wonder whether the cardiomyopathy has been described in other varieties other than Malinois. If not, I would suggest including Malinois in the title. 
  4. Line 76 - misspelling of "family"
  5. Line 80 - as I read this sentence, I would consider removing the first "or" before "missing in any of..." The sentence reads as a list, and having two "or" connectors interrupts the list. 
  6. Section 2.6: how was the affected Malinois selected for sequencing? Was the dog selected before or after mapping? I am interested in whether you selected a sample based on homozygosity. 
  7. I am not a veterinarian, so my comments to the phenotype description may be basic. I do question the word choice in line 124: "minimal chronic lymphocytic myocarditis" seems contradictory? The puppies were 8 weeks old - could they have had chronic myocarditis? Lymphocytic myocarditis is often associated with a viral infection. Was any viral infection noted or anticipated? Given the cardiac involvement, was any arrhythmia noted among affected dogs prior to death?
  8. Lines 129-136 were more vague than I would have liked. In describing the third puppy, the timeline to death is confusing. As I read the paragraph, an 8-week old puppy died suddenly after two weeks of clinical signs and treatment? Is this correct? Line 134 needs detail regarding the "mild changes" (please describe the changes, however mild, as you are characterizing a phenotype) and "consistent with an ongoing inflammatory processes" has not been introduced or suggested from other data previously provided. 
  9. Figure 1 legend: edit text for (A) to read "Cardiomyocytes of an affected puppy are swollen and pale, and the sarcoplasm..." 
  10. Section 3.2: Why was the third affected puppy not included in linkage/homozygosity mapping to narrow regions of IBD/IBS?
  11. From a visual perspective, I think I would like the column header "Filtering step" for Table 1 to be left justified, rather than centered on column. But that's a personal preference.
  12. Table 2: I am interested in seeing an additional column with a predicted impact score from SnpEff or other predictive program. Predicted score is not provided in results anywhere.
  13. The paragraph extending lines 203-216 needs more references cited throughout for the information provided.
  14. Line 213 - Please check Schizosachormyces for spelling (Schizosaccharomyces pombe).
  15. Line 225 and 229 - use "intermittent" instead of "intermitted."

I would love to read a discussion point describing the DLA genotypes of affected puppies to know what MHC alleles may contribute to susceptibility, or different phenotype presentations. I know you couldn't make any associations with only three affecteds, but would like to see MHC, immunity, and MLASA analog discussed. 

Given the high carrier rate in Belgian Shepherds, is it possible that affected pups could be homozygous lethal in severe manifestation? 

Author Response

(1)

The sentence spanning lines 41-43 needs some attention. "fatal diseases" and "puppy loss" seem redundant, as "fatal" would indicate loss of puppy. I would suggest the sentence read "...lead to fatal diseases, or predispose for secondary infectious diseases [3]." As written with the "and" suggests secondary diseases could occur after puppy loss.

Response: Revised accordingly.

(2)

I would like line 50 to indicate the number of dogs cited, rather than "several" - is this sentence referring to the three affected dogs included in the study, or are there more families with puppy loss?

Response: We changed this to: “breeders noticed three puppies of the Malinois variety dying from unknown causes”.

(3)

I was not aware the Belgian Shepherd was described as 4 AKC breeds and wonder whether the cardiomyopathy has been described in other varieties other than Malinois. If not, I would suggest including Malinois in the title.

Response: As this is a European study and a European journal, we kept the European (FCI) breed nomenclature and did not change the title. The breed nomenclature is explained in section 2.2 in the methods. The four different varieties may be intercrossed. As the presumed disease allele is present in at least three of the four varieties, we do not consider it appropriate to limit the title to the Malinois variety. We added the four varieties of Belgian Shepherds to the keywords.

(4)

Line 76 - misspelling of "family"

Response: Revised accordingly.

(5)

Line 80 - as I read this sentence, I would consider removing the first "or" before "missing in any of..." The sentence reads as a list, and having two "or" connectors interrupts the list.

Response: Revised accordingly.

(6)

Section 2.6: how was the affected Malinois selected for sequencing? Was the dog selected before or after mapping? I am interested in whether you selected a sample based on homozygosity.

Response: For practical reasons, we initiated the mapping experiment and the whole genome sequencing experiment simultaneously. At the time when we selected the sample for sequencing, we were not yet aware of the third case from Germany. We selected the female sibling of the two initially available Swiss cases for sequencing, in order to have two copies of the X‑chromosome. This choice will make the dog genome more useful as a control for future projects. As this consideration is not relevant for the present study, we did not change the text with respect to the comment.

In our opinion, it is not necessary to select the dog with the smallest homozygous segment for whole genome sequencing. Once the minimally shared homozygous segment is defined, one can restrict the search for potential causative variants to this interval in the genome sequence of the actually sequenced individual case.

(7)

I am not a veterinarian, so my comments to the phenotype description may be basic. I do question the word choice in line 124: "minimal chronic lymphocytic myocarditis" seems contradictory? The puppies were 8 weeks old - could they have had chronic myocarditis? Lymphocytic myocarditis is often associated with a viral infection. Was any viral infection noted or anticipated? Given the cardiac involvement, was any arrhythmia noted among affected dogs prior to death?

Response: In pathology a chronic inflammation is an inflammation of prolonged duration (weeks to months). Inflammatory cells associated with a chronic inflammation are lymphocytes, among other inflammatory cells. The ongoing accumulation of material and probable disturbance of the mitochondria and cardiomyocyte metabolism led to degeneration and apoptosis/ necrosis of the cardiomyocytes. This then elicits an inflammatory process leading to the lymphocytic infiltration. This process was ongoing for several weeks and is thus defined as a chronic pathologic process.

As the reviewer correctly stated, viral infections of the heart cause a lymphocytic inflammation. We therefore checked for a canine parvoviral infection, which can cause a lymphocytic myocarditis especially in that age group of dogs. The results were negative. As the liver in all three puppies and the brain and lung in the two sibling puppies were of normal appearance, no further viral testing for Canine Adenovirus 1 and Canine Distemper Virus were performed. These two viral infections are the most likely infectious differentials for canine puppies of this age group in Europe. In the third puppy, the clinical signs of myelitis and the distribution and presentation of the inflammation in the central nervous system are highly unlikely for a Canine Distemper Virus infection. The serum of the affected puppy was checked for antibodies against Neospora caninum and Toxoplasma gondii, other possible causes for the clinical signs. The results were negative and in histology no indication of such a protozoal infection was found.

In line 128 we added: There was a minimal chronic lymphocytic myocarditis as a result of the degeneration of affected cardiomyocytes.

(8)

Lines 129-136 were more vague than I would have liked. In describing the third puppy, the timeline to death is confusing. As I read the paragraph, an 8-week old puppy died suddenly after two weeks of clinical signs and treatment? Is this correct? Line 134 needs detail regarding the "mild changes" (please describe the changes, however mild, as you are characterizing a phenotype) and "consistent with an ongoing inflammatory processes" has not been introduced or suggested from other data previously provided.

Response: We brought the timeline into a chronological order and added more details concerning the changes in the spinal cord and brain.

Lines 134 – 141 now read: A third male puppy became reluctant to move while twitching and showing tremors at six weeks of age. During these bouts of tremors and twitching it exhibited spontaneous urination and defecation. The puppy was treated for myelitis and died suddenly at 8 weeks of age. The necropsy and histologic examination revealed similar changes in the heart as with the other two puppies where cardiomyocytes appeared to have a grainy sarcoplasm. Furthermore, there were mild changes in the brain and spinal cord with few chromatolytic neurons in the spinal cord, mild gliosis in the cerebellum and a mild mixed cell infiltration of the meninges of the cerebellum. Additionally, the puppy had a catarrhal enteritis, for which the cause could not be determined.

(9)

Figure 1 legend: edit text for (A) to read "Cardiomyocytes of an affected puppy are swollen and pale, and the sarcoplasm..."

Response: Revised accordingly.

(10)

Section 3.2: Why was the third affected puppy not included in linkage/homozygosity mapping to narrow regions of IBD/IBS?

Response: We started this investigation with the 2 Swiss cases from Switzerland and identified the candidate causative variant using only this small family (grey rectangle in Figure 2). Only when we genotyped our entire collection of Belgian Shepherd samples, we realized that we had a third homozygous mutant dog from Germany in our cohort. We then checked the pedigree of this dog and discovered that it was related to the two Swiss cases. Finally, we re-evaluated the archived FFPE samples of the German case and discovered that it shared many phenotypic similarities with the two Swiss cases.

We think that this sequence of events actually strengthens our claim of causality for the YARS2:c1054G>A variant as we consider it very unlikely that a dog that shows a rare homozygous genotype by chance dies at exactly the same age as the two initial cases and also showed a comparable histopathologic phenotype of the heart. The clinical course of the third case prior to death was quite distinct from the two Swiss cases, which is a typical feature of this phenotype.

We chose to present the manuscript in a logical order starting with the phenotypic description of all three cases as we thought that a narrative of the true historic order of our research project might be quite confusing to the reader.

(11)

From a visual perspective, I think I would like the column header "Filtering step" for Table 1 to be left justified, rather than centered on column. But that's a personal preference.

Response: We completely agree with the reviewer and set the header left justified. However, this may change again due to journal style.

(12)

Table 2: I am interested in seeing an additional column with a predicted impact score from SnpEff or other predictive program. Predicted score is not provided in results anywhere.

Response: SnpEff classifications are given in Table S4 for all private variants (effect and impact). However, SnpEff does not make predictions about the potential pathogenicity of different missense variants. According to our own experience, in silico predictions of the functional impact or pathogenicity of missense variants have a relatively low reliability. We consider it much more meaningful to present the evolutionary conservation (Figure 3B) and details on the mutated residue in the three-dimensional structure of the protein (Figure 3C and D).

To address the reviewer’s comment we added the predictions of Sift and PMut to the results section. Both software tools predict that the variant affects protein function.

(13)

The paragraph extending lines 203-216 needs more references cited throughout for the information provided.

Response: We added one new reference and inserted several additional citations.

(14)

Line 213 - Please check Schizosachormyces for spelling (Schizosaccharomyces pombe).

Response: Revised accordingly.

(15)

Line 225 and 229 - use "intermittent" instead of "intermitted."

Response: Revised accordingly.

(16)

I would love to read a discussion point describing the DLA genotypes of affected puppies to know what MHC alleles may contribute to susceptibility, or different phenotype presentations. I know you couldn't make any associations with only three affecteds, but would like to see MHC, immunity, and MLASA analog discussed.

Response: We appreciate the reviewer’s comment. It would indeed be very interesting to investigate whether the DLA genotype modulates the specific disease phenotype in these dogs. Unfortunately, as we are not really experts in immunogenetics, we were not able to add a truly meaningful paragraph on this aspect to the discussion.

(17)

Given the high carrier rate in Belgian Shepherds, is it possible that affected pups could be homozygous lethal in severe manifestation?

Response: We do not understand this question. All three puppies with the homozygous mutant genotype died at a very young age (= lethal). Was the reviewer intending to ask whether pre-natal or perinatal lethality might be caused by the homozygous genotype? We don’t have experimental data to answer this question and did not want to expand the manuscript with unfounded speculations.